# Online Prediction of Switching Graph Labelings with Cluster Specialists

**Mark Herbster**
Department of Computer Science
University College London
London
United Kingdom
m.herbster@cs.ucl.ac.uk

**James Robinson**
Department of Computer Science
University College London
London
United Kingdom
j.robinson@cs.ucl.ac.uk

## Abstract

We address the problem of predicting the labeling of a graph in an online setting
when the labeling is changing over time. We present an algorithm based on a
*specialist* [11] approach; we develop the machinery of cluster specialists which
probabilistically exploits the cluster structure in the graph. Our algorithm has
two variants, one of which surprisingly only requires $\mathcal{O}(\log n)$ time on any trial
$t$ on an $n$-vertex graph, an exponential speed up over existing methods. We
prove switching mistake-bound guarantees for both variants of our algorithm.
Furthermore these mistake bounds *smoothly* vary with the magnitude of the change
between successive labelings. We perform experiments on Chicago Divvy Bicycle
Sharing data and show that our algorithms significantly outperform an existing
algorithm (a kernelized Perceptron) as well as several natural benchmarks.

## 1 Introduction

We study the problem of predicting graph labelings that evolve over time. Consider the following
game for predicting the labeling of a graph in the online setting. `Nature` presents a graph $\mathcal{G}$; `Nature`
queries a vertex $i_1 \in V = \{1, 2, \ldots, n\}$; the `learner` predicts the label of the vertex $\hat{y}_1 \in \{-1, 1\}$;
`Nature` presents a label $y_1$; `Nature` queries a vertex $i_2$; the `learner` predicts $\hat{y}_2$; and so forth. The
`learner`'s goal is to minimize the total number of mistakes $M = |\{t : \hat{y}_t \neq y_t\}|$. If `Nature` is
strictly adversarial, the `learner` will incur a mistake on every trial, but if `Nature` is regular or
simple, there is hope that the `learner` may incur only a few mistakes. Thus, a central goal of
mistake-bounded online learning is to design algorithms whose total mistakes can be bounded relative
to the complexity of `Nature`'s labeling. This (non-switching) graph labeling problem has been
studied extensively in the online learning literature [16, 15, 7, 34, 17]. In this paper we generalize
the setting to allow the underlying labeling to change arbitrarily over time. The `learner` has no
knowledge of when a change in labeling will occur and therefore must be able to adapt quickly to
these changes.

Consider an example of services placed throughout a city, such as public bicycle sharing stations.
As the population uses these services the state of each station–such as the number of available
bikes–naturally evolves throughout the day, at times gradually and others abruptly, and we might
want to predict the state of any given station at any given time. Since the location of a given station
as well as the state of nearby stations will be relevant to this learning problem it is natural to use a
graph-based approach. Another setting might be a graph of major road junctions (vertices) connected
by roads (edges), in which one wants to predict whether or not a junction is congested at any given
time. Traffic congestion is naturally non-stationary and also exhibits both gradual and abrupt changes
to the structure of the labeling over time [24].

The structure of this paper is as follows. In Section 2 we discuss the background literature. In Section 3 we present the SWITCHING CLUSTER SPECIALISTS algorithm (SCS), a modification of the method of specialists [11] with the novel machinery of *cluster specialists*, a set of specialists that in a rough sense correspond to clusters in the graph. We consider two distinct sets of specialists, $\mathcal{B}_n$ and $\mathcal{F}_n$, where $\mathcal{B}_n \subset \mathcal{F}_n$. With the smaller set of specialists the bound is only larger by factor of $\log n$. On the other hand, prediction is exponentially faster per trial, remarkably requiring only $\mathcal{O}(\log n)$ time to predict. In Section 4 we provide experiments on Chicago Divvy Bicycle Sharing data. In Section 5 we provide some concluding remarks. All proofs are contained in the technical appendices.

## 1.1 Notation

We first present common notation. Let $\mathcal{G} = (V, E)$ be an undirected, connected, $n$-vertex graph with vertex set $V = \{1, 2, \ldots, n\}$ and edge set $E$. Each vertex of this graph may be labeled with one of two states $\{-1, 1\}$ and thus a labeling of a graph may be denoted by a vector $\boldsymbol{u} \in \{-1, 1\}^n$ where $u_i$ denotes the label of vertex $i$. The underlying assumption is that we are predicting vertex labels from a sequence $\boldsymbol{u}_1, \ldots, \boldsymbol{u}_T \in \{-1, 1\}^n$ of graph labelings over $T$ trials. The set $K := \{t \in \{2, \ldots, T\} : \boldsymbol{u}_t \neq \boldsymbol{u}_{t-1}\} \cup \{1\}$ contains the first trial of each of the $|K|$ "segments" of the prediction problem. Each segment corresponds to a time period when the underlying labeling is unchanging. The *cut-size* of a labeling $\boldsymbol{u}$ on a graph $\mathcal{G}$ is defined as $\Phi_{\mathcal{G}}(\boldsymbol{u}) := |\{(i, j) \in E : u_i \neq u_j\}|$, i.e., the number of edges between vertices of disagreeing labels.

We let $r_{\mathcal{G}}(i, j)$ denote the *resistance distance* (effective resistance) between vertices $i$ and $j$ when the graph $\mathcal{G}$ is seen as a circuit where each edge has unit resistance (e.g., [26]). The effective resistance for an unweighted graph $\mathcal{G}$ can be written as

$$r_{\mathcal{G}}(i, j) = \frac{1}{\min\limits_{\boldsymbol{u} \in \mathbb{R}^n} \sum\limits_{(p,q) \in E} (u_p - u_q)^2 : u_i - u_j = 1}$$

The *resistance diameter* of a graph is $R_{\mathcal{G}} := \max\limits_{i,j \in V} r_{\mathcal{G}}(i, j)$. The *resistance weighted* cut-size of a labeling $\boldsymbol{u}$ is $\Phi_{\mathcal{G}}^r(\boldsymbol{u}) := \sum\limits_{(i,j) \in E : u_i \neq u_j} r_{\mathcal{G}}(i, j)$. Let $\Delta_n = \{\boldsymbol{\mu} \in [0, 1]^n : \sum_{i=1}^n \mu_i = 1\}$ be the $n$-dimensional probability simplex. For $\boldsymbol{\mu} \in \Delta_n$ we define $H(\boldsymbol{\mu}) := \sum_{i=1}^n \mu_i \log_2 \frac{1}{\mu_i}$ to be the entropy of $\boldsymbol{\mu}$. For $\boldsymbol{\mu}, \boldsymbol{\omega} \in \Delta_n$ we define $d(\boldsymbol{\mu}, \boldsymbol{\omega}) = \sum_{i=1}^n \mu_i \log_2 \frac{\mu_i}{\omega_i}$ to be the relative entropy between $\boldsymbol{\mu}$ and $\boldsymbol{\omega}$. For a vector $\boldsymbol{\omega}$ and a set of indices $\mathcal{I}$ let $\boldsymbol{\omega}(\mathcal{I}) := \sum_{i \in \mathcal{I}} \omega_i$. For any positive integer $N$ we define $[N] := \{1, 2, \ldots, N\}$ and for any predicate $[\text{PRED}] := 1$ if PRED is true and equals 0 otherwise.

## 2 Related Work

The problem of predicting the labeling of a graph in the batch setting was introduced as a foundational method for semi-supervised (transductive) learning. In this work, the graph was built using both the unlabeled and labeled instances. The seminal work by [3] used a metric on the instance space and then built a kNN or $\epsilon$-ball graph. The partial labeling was then extended to the complete graph by solving a mincut-maxflow problem where opposing binary labels represented sources and sinks. In practice this method suffered from very unbalanced cuts. Significant practical and theoretical advances were made by replacing the mincut/maxflow model with methods based on minimising a quadratic form of the graph Laplacian. Influential early results include but are not limited to [39, 2, 38]. A limitation of the graph Laplacian-based techniques is that these batch methods–depending on their implementation–typically require $\Theta(n^2)$ to $\Theta(n^3)$ time to produce a single set of predictions.

Predicting the labeling of a graph in the online setting was introduced by [20]. The authors proved bounds for a Perceptron-like algorithm with a kernel based on the graph Laplacian. Since this work there has been a number of extensions and improvements in bounds including but not limited to [16, 6, 15, 18, 17, 32]. Common to all of these papers is that a dominant term in their mistake bounds is the (resistance-weighted) cut-size.

From a simplified perspective, the methods for predicting the labeling of a graph (online) split into two approaches. The first approach works directly with the original graph and is usually based on a graph Laplacian [20, 15, 17]; it provides bounds that utilize the additional connectivity of non-tree graphs, which are particularly strong when the graph contains uniformly-labeled clusters of small

(resistance) diameter. The drawbacks of this approach are that the bounds are weaker on graphs with large diameter, and that computation times are slower.

The second approach is to approximate the original graph with an appropriately selected tree or "line" graph [16, 7, 6, 34]. This enables faster computation times, and bounds that are better on graphs with large diameters. These algorithms may be extended to non-tree graphs by first selecting a spanning tree uniformly at random [7] and then applying the algorithm to the sampled tree. This randomized approach induces *expected* mistake bounds that also exploit the cluster structure in the graph (see Section 2.2). Our algorithm takes this approach.

## 2.1 Switching Prediction

In this paper rather than predicting a single labeling of a graph we instead will predict a (switching) sequence of labelings. *Switching* in the mistake- or regret-bound setting refers to the problem of predicting an online sequence when the "best comparator" is changing over time. In the simplest of switching models the set of comparators is *structureless* and we simply pay per switch. A prominent early result in this model is [21] which introduced the *fixed-share* update which will play a prominent role in our main algorithm. Other prominent results in the structureless model include but are not limited to [36, 4, 12, 28, 27, 5]. A stronger model is to instead prove a bound that holds for any arbitrary contiguous sequence of trials. Such a bound is called an *adaptive-regret* bound. This type of bound automatically implies a bound on the structureless switching model. Adaptive-regret was introduced in [13][1] other prominent results in this model include [1, 5, 9].

The structureless model may be generalized by introducing a divergence measure on the set of comparators. Thus, whereas in the structureless model we pay for the number of switches, in the structured model we instead pay in the sum of divergences between successive comparators. This model was introduced in [22]; prominent results include [25, 5].

In [12, 23, 13] meta-algorithms were introduced with regret bounds which convert any "black-box" online learning algorithm into an adaptive algorithm. Such methods could be used as an approach to predict switching graph labelings online, however these meta-algorithms introduce a factor of $\mathcal{O}(\log T)$ to the per-trial time complexity of the base online learning algorithm. In the online switching setting we will aim for our fastest algorithm to have $\mathcal{O}(\log n)$ time complexity per trial.

In [18] the authors also consider switching graph label prediction. However, their results are not directly comparable to ours since they consider the combinatorially more challenging problem of repeated switching within a small set of labelings contained in a larger set. That set-up was a problem originally framed in the "experts" setting and posed as an open problem by [10] and solved in [4]. If we apply the bound in [18] to the case where there is *not* repeated switching within a smaller set, then their bound is uniformly and significantly weaker than the bounds in this paper and the algorithm is quite slow requiring $\theta(n^3)$ time per trial in a typical implementation. Also contained in [18] is a baseline algorithm based on a kernel perceptron with a graph Laplacian kernel. The bound of that algorithm has the significant drawback in that it scales with respect to the "worst" labeling in a sequence of labelings. However, it is simple to implement and we use it as a benchmark in our experiments.

## 2.2 Random Spanning Trees and Linearization

Since we operate in the transductive setting where the entire unlabeled graph is presented to the `learner` beforehand, this affords the `learner` the ability to perform any reconfiguration to the graph as a preprocessing step. The bounds of most existing algorithms for predicting a labeling on a graph are usually expressed in terms of the cut-size of the graph under that labeling. A natural approach then is to use a spanning tree of the original graph which can only reduce the cut-size of the labeling.

The effective resistance between vertices $i$ and $j$, denoted $r_{\mathcal{G}}(i, j)$, is equal to the probability that a spanning tree of $\mathcal{G}$ drawn uniformly at random (from the set of all spanning trees of $\mathcal{G}$) includes $(i, j) \in E$ as one of its $n - 1$ edges (e.g., [30]). As first observed by [6], by selecting a spanning tree uniformly at random from the set of all possible spanning trees, mistake bounds expressed in terms of the cut-size then become *expected* mistake bounds now in terms of the effective-resistance-weighted cut-size of the graph. That is, if $\mathcal{R}$ is a random spanning tree of $\mathcal{G}$ then $\mathbb{E}[\Phi_{\mathcal{R}}(\boldsymbol{u})] = \Phi_{\mathcal{G}}^r(\boldsymbol{u})$ and thus

$\Phi_{\mathcal{G}}^r(\boldsymbol{u}) \leq \Phi_{\mathcal{G}}(\boldsymbol{u})$. A random spanning tree can be sampled from a graph efficiently using a random walk or similar methods (see e.g., [37]).

To illustrate the power of this randomization consider the simplified example of a graph with two cliques each of size $\frac{n}{2}$, where one clique is labeled uniformly with '+1' and the other '-1' with an additional arbitrary $\frac{n}{2}$ "cut" edges between the cliques. This dense graph exhibits two disjoint clusters and $\Phi_{\mathcal{G}}(\boldsymbol{u}) = \frac{n}{2}$. On the other hand $\Phi_{\mathcal{G}}^r(\boldsymbol{u}) = \Theta(1)$, since between any two vertices in the opposing cliques there are $\frac{n}{2}$ edge disjoint paths of length $\leq 3$ and thus the effective resistance between any pair of vertices is $\Theta(\frac{1}{n})$. Since bounds usually scale linearly with (resistance-weighted) cut-size, the cut-size bound would be vacuous but the resistance-weighted cut-size bound would be small.

We will make use of this preprocessing step of sampling a uniform random spanning tree, as well as a *linearization* of this tree to produce a (spine) line-graph, $\mathcal{S}$. The linearization of $\mathcal{G}$ to $\mathcal{S}$ as a preprocessing step was first proposed by [16] and has since been applied in, e.g., [7, 31]. In order to construct $\mathcal{S}$, a random-spanning tree $\mathcal{R}$ is picked uniformly at random. A vertex of $\mathcal{R}$ is then chosen and the graph is fully traversed using a *depth-first search* generating an ordered list $V_{\mathcal{L}} = \left\{ i_{l_1}, \ldots, i_{l_{2m+1}} \right\}$ of vertices in the order they were visited. Vertices in $V$ may appear multiple times in $V_{\mathcal{L}}$. A subsequence $V_{\mathcal{L}'} \subseteq V_{\mathcal{L}}$ is then chosen such that each vertex in $V$ appears only once. The line graph $\mathcal{S}$ is then formed by connecting each vertex in $V_{\mathcal{L}'}$ to its immediate neighbors in $V_{\mathcal{L}'}$ with an edge. We denote the edge set of $\mathcal{S}$ by $E_{\mathcal{S}}$ and let $\Phi_t := \Phi(\boldsymbol{u}_t)$, where the cut $\Phi$ is with respect to the linear embedding $\mathcal{S}$. Surprisingly, as stated in the lemma below, the cut on this linearized graph is no more than twice the cut on the original graph.

**Lemma 1** ([16]). *Given a labeling $\boldsymbol{u} \in \{-1, 1\}^n$ on a graph $\mathcal{G}$, for the mapping $\mathcal{G} \to \mathcal{R} \to \mathcal{S}$, as above, we have $\Phi_{\mathcal{S}}(\boldsymbol{u}) \leq 2\Phi_{\mathcal{R}}(\boldsymbol{u}) \leq 2\Phi_{\mathcal{G}}(\boldsymbol{u})$.*

By combining the above observations we may reduce the problem of learning on a graph to that of learning on a line graph. In particular, if we have an algorithm with a mistake bound of the form $M \leq \mathcal{O}(\Phi_{\mathcal{G}}(\boldsymbol{u}))$ this implies we then may give an *expected* mistake bound of the form $M \leq \mathcal{O}(\Phi_{\mathcal{G}}^r(\boldsymbol{u}))$ by first sampling a random spanning tree and then linearizing it as above. One caveat of this however depends on the whether `Nature` is *oblivious* or *adaptive*. If `Nature` is oblivious we assume that `learner`'s predictions have no effect on the labels chosen by `Nature` (or equivalently all labelings are chosen beforehand). Conversely if `Nature` is adaptive then `Nature`'s labelings are assumed to be adversarially chosen in response to `learner`'s predictions. In this paper we will only state the deterministic mistake bounds in terms of cut-size which will hold for oblivious and adaptive adversaries, while the expected bounds in terms of resistance-weighted cut-sizes will hold for an oblivious adversary.

## 3 Switching Specialists

In this section we present a new method based on the idea of *specialists* [11] from the *prediction with expert advice* literature [29, 35, 8]. Although the achieved bounds are slightly worse than other methods for predicting a *single* labeling of a graph, the derived advantage is that it is possible to obtain "competitive" bounds with fast algorithms to predict a sequence of changing graph labelings.

Our inductive bias is to predict well when a labeling has a *small* (resistance-weighted) cut-size. The complementary perspective implies that the labeling consists of a *few* uniformly labeled clusters. This suggests the idea of maintaining a collection of basis functions where each such function is specialized to predict a constant function on a given cluster of vertices. To accomplish this technically we adapt the method of *specialists* [11, 27]. A specialist is a prediction function $\varepsilon$ from an input space to an extended output space with *abstentions*. So for us the input space is just $V = [n]$, the vertices of a graph; and the extended output space is $\{-1, 1, \square\}$ where $\{-1, 1\}$ corresponds to predicted labels of the vertices, but '$\square$' indicates that the specialist abstains from predicting. Thus a specialist *specializes* its prediction to part of the input space and in our application the specialists correspond to a collection of clusters which cover the graph, each cluster uniformly predicting $-1$ or $1$.

In Algorithm 1 we give our switching specialists method. The algorithm maintains a weight vector $\boldsymbol{\omega}_t$ over the specialists in which the magnitudes may be interpreted as the current confidence we have in each of the specialists. The updates and their analyses are a combination of three standard methods: i) *Halving* loss updates, ii) specialists updates and iii) (delayed) fixed-share updates.

```
input        : Specialists set $\mathcal{E}$
parameter : $\alpha \in [0,1]$
initialize  : $\boldsymbol{\omega}_1 \leftarrow \frac{1}{|\mathcal{E}|}\mathbf{1}, \dot{\boldsymbol{\omega}}_0 \leftarrow \frac{1}{|\mathcal{E}|}\mathbf{1}, \boldsymbol{p} \leftarrow \mathbf{0}, m \leftarrow 0$
for $t = 1$ to $T$ do
    receive $i_t \in V$
    set $\mathcal{A}_t := \{\varepsilon \in \mathcal{E} : \varepsilon(i_t) \neq \square\}$
    foreach $\varepsilon \in \mathcal{A}_t$ do                                    // delayed share update
        $\omega_{t,\varepsilon} \leftarrow (1-\alpha)^{m-p_\varepsilon}\dot{\omega}_{t-1,\varepsilon} + \frac{1-(1-\alpha)^{m-p_\varepsilon}}{|\mathcal{E}|}$        (1)

    predict $\hat{y}_t \leftarrow \text{sign}(\sum_{\varepsilon \in \mathcal{A}_t} \omega_{t,\varepsilon}\,\varepsilon(i_t))$
    receive $y_t \in \{-1, 1\}$
    set $\mathcal{Y}_t := \{\varepsilon \in \mathcal{E} : \varepsilon(i_t) = y_t\}$
    if $\hat{y}_t \neq y_t$ then                                                  // loss update
        $\dot{\omega}_{t,\varepsilon} \leftarrow \begin{cases} 0 & \varepsilon \in \mathcal{A}_t \cap \bar{\mathcal{Y}}_t \\ \dot{\omega}_{t-1,\varepsilon} & \varepsilon \notin \mathcal{A}_t \\ \omega_{t,\varepsilon}\frac{\boldsymbol{\omega}_t(\mathcal{A}_t)}{\boldsymbol{\omega}_t(\mathcal{Y}_t)} & \varepsilon \in \mathcal{Y}_t \end{cases}$        (2)

        foreach $\varepsilon \in \mathcal{A}_t$ do
            $p_\varepsilon \leftarrow m$
        $m \leftarrow m + 1$
    else
        $\dot{\boldsymbol{\omega}}_t \leftarrow \dot{\boldsymbol{\omega}}_{t-1}$
```

**Algorithm 1:** SWITCHING CLUSTER SPECIALISTS

The loss update (2) zeros the weight components of incorrectly predicting specialists, while the non-predicting specialists are not updated at all. In (1) we give our *delayed* fixed-share style update.

A standard fixed share update may be written in the following form:

$$\omega_{t,\varepsilon} = (1-\alpha)\dot{\omega}_{t-1,\varepsilon} + \frac{\alpha}{|\mathcal{E}|}\,. \qquad (3)$$

Although (3) superficially appears different to (1), in fact these two updates are exactly the same in terms of predictions generated by the algorithm. This is because (1) caches updates until the given specialist is again active. The purpose of this computationally is that if the active specialists are, for example, logarithmic in size compared to the total specialist pool, we may then achieve an exponential speedup over (3); which in fact we will exploit.

In the following theorem we will give our switching specialists bound. The dominant cost of switching on trial $t$ to $t+1$ is given by the non-symmetric $J_\mathcal{E}(\boldsymbol{\mu}_t, \boldsymbol{\mu}_{t+1}) := |\{\varepsilon \in \mathcal{E} : \mu_{t,\varepsilon} = 0, \mu_{t+1,\varepsilon} \neq 0\}|$, i.e., we pay only for each new specialist introduced but we do not pay for removing specialists.

**Theorem 2.** *For a given specialist set $\mathcal{E}$, let $M_\mathcal{E}$ denote the number of mistakes made in predicting the online sequence $(i_1, y_1), \ldots, (i_T, y_T)$ by Algorithm 1. Then,*

$$M_\mathcal{E} \leq \frac{1}{\pi_1}\log|\mathcal{E}| + \sum_{t=1}^{T}\frac{1}{\pi_t}\log\frac{1}{1-\alpha} + \sum_{i=1}^{|K|-1}J_\mathcal{E}\big(\boldsymbol{\mu}_{k_i}, \boldsymbol{\mu}_{k_{i+1}}\big)\log\frac{|\mathcal{E}|}{\alpha}\,, \qquad (4)$$

*for any sequence of **consistent** and **well-formed** comparators $\boldsymbol{\mu}_1, \ldots, \boldsymbol{\mu}_T \in \Delta_{|\mathcal{E}|}$ where $K := \{k_1 = 1 < \cdots < k_{|K|}\} := \{t \in [T] : \boldsymbol{\mu}_t \neq \boldsymbol{\mu}_{t-1}\} \cup \{1\}$, and $\pi_t := \boldsymbol{\mu}_t(\mathcal{Y}_t)$.*

The bound in the above theorem depends crucially on the best sequence of *consistent* and *well-formed* comparators $\boldsymbol{\mu}_1, \ldots, \boldsymbol{\mu}_T$. The consistency requirement implies that on every trial there is no active incorrect specialist assigned "mass" ($\boldsymbol{\mu}_t(\mathcal{A}_t \setminus \mathcal{Y}_t) = 0$). We may eliminate the consistency requirement by "softening" the loss update (2). A comparator $\boldsymbol{\mu} \in \Delta_{|\mathcal{E}|}$ is *well-formed* if $\forall\, v \in V$, there exists a *unique* $\varepsilon \in \mathcal{E}$ such that $\varepsilon(v) \neq \square$ and $\mu_\varepsilon > 0$, and furthermore there exists a $\pi \in (0, 1]$ such that $\forall \varepsilon \in \mathcal{E} : \mu_\varepsilon \in \{0, \pi\}$, i.e., each specialist in the support of $\boldsymbol{\mu}$ has the same mass $\pi$ and these specialists disjointly cover the input space ($V$). At considerable complication to the form of the bound the well-formedness requirement may be eliminated.

The above bound is "smooth" in that it scales with a gradual change in the comparator. In the next section we describe the novel specialists sets that we've tailored to graph-label prediction so that a small change in comparator corresponds to a small change in a graph labeling.

### 3.1 Cluster Specialists

In order to construct the *cluster specialists* over a graph $\mathcal{G} = (V = [n], E)$, we first construct a line graph as described in Section 2.2. A cluster specialist is then defined by $\varepsilon_y^{l,r}(\cdot)$ which maps $V \to \{-1, 1, \square\}$ where $\varepsilon_y^{l,r}(v) := y$ if $l \leq v \leq r$ and $\varepsilon_y^{l,r}(v) := \square$ otherwise. Hence cluster specialist $\varepsilon_y^{l,r}(v)$ corresponds to a function that predicts the label $y$ if vertex $v$ lies between vertices $l$ and $r$ and abstains otherwise. Recall that by sampling a random spanning tree the expected cut-size of a labeling on the spine is no more than twice the resistance-weighted cut-size on $\mathcal{G}$. Thus, given a labeled graph with a small resistance-weighted cut-size with densely interconnected clusters and modest intra-cluster connections, this implies a cut-bracketed linear segment on the spine will in expectation roughly correspond to one of the original dense clusters. We will consider two basis sets of cluster specialists.

**Basis $\mathcal{F}_n$:** We first introduce the *complete* basis set $\mathcal{F}_n := \{\varepsilon_y^{l,r} : l, r \in [n], l \leq r; y \in \{-1, 1\}\}$. We say that a set of specialists $\mathcal{C}_{\boldsymbol{u}} \subseteq \mathcal{E} \subseteq 2^{\{-1,1,\square\}^n}$ from basis $\mathcal{E}$ *covers* a labeling $\boldsymbol{u} \in \{-1, 1\}^n$ if for all $v \in V = [n]$ and $\varepsilon \in \mathcal{C}_{\boldsymbol{u}}$ that $\varepsilon(v) \in \{u_v, \square\}$ and if $v \in V$ then there exists $\varepsilon \in \mathcal{C}_{\boldsymbol{u}}$ such that $\varepsilon(v) = u_v$. The basis $\mathcal{E}$ is *complete* if every labeling $\boldsymbol{u} \in \{-1, 1\}^n$ is covered by some $\mathcal{C}_{\boldsymbol{u}} \subseteq \mathcal{E}$. The basis $\mathcal{F}_n$ is complete and in fact has the following approximation property: for any $\boldsymbol{u} \in \{-1, 1\}^n$ there exists a covering set $\mathcal{C}_{\boldsymbol{u}} \subseteq \mathcal{F}_n$ such that $|\mathcal{C}_{\boldsymbol{u}}| = \Phi_{\mathcal{S}}(\boldsymbol{u}) + 1$. This follows directly as a line with $k - 1$ cuts is divided into $k$ segments. We now illustrate the use of basis $\mathcal{F}_n$ to predict the labeling of a graph. For simplicity we illustrate by considering the problem of predicting a single graph labeling without switching. As there is no switch we will set $\alpha := 0$ and thus if the graph is labeled with $\boldsymbol{u} \in \{-1, 1\}^n$ with cut-size $\Phi_{\mathcal{S}}(\boldsymbol{u})$ then we will need $\Phi_{\mathcal{S}}(\boldsymbol{u}) + 1$ specialists to predict the labeling and thus the comparators may be post-hoc optimally determined so that $\boldsymbol{\mu} = \boldsymbol{\mu}_1 = \cdots = \boldsymbol{\mu}_T$ and there will be $\Phi_{\mathcal{S}}(\boldsymbol{u}) + 1$ components of $\boldsymbol{\mu}$ each with "weight" $\frac{1}{(\Phi_{\mathcal{S}}(\boldsymbol{u})+1)}$, thus $\frac{1}{\pi_1} = \Phi_{\mathcal{S}}(\boldsymbol{u}) + 1$, since there will be only one specialist (with non-zero weight) active per trial. Since the cardinality of $\mathcal{F}_n$ is $n^2 + n$, by substituting into (4) we have that the number of mistakes will be bounded by $(\Phi_{\mathcal{S}}(\boldsymbol{u}) + 1) \log (n^2 + n)$. Note for a single graph labeling on a spine this bound is not much worse than the best known result [16, Theorem 4]. In terms of computation time however it is significantly slower than the algorithm in [16] requiring $\Theta(n^2)$ time to predict on a typical trial since on average there are $\Theta(n^2)$ specialists active per trial.

**Basis $\mathcal{B}_{1,n}$:** We now introduce the basis $\mathcal{B}_n$ which has $\Theta(n)$ specialists and only requires $\mathcal{O}(\log n)$ time per trial to predict with only a small increase in bound. The basis is defined as

$$\mathcal{B}_{p,q} := \begin{cases} \{\varepsilon_{-1}^{p,q}, \varepsilon_1^{p,q}\} & p = q, \\ \{\varepsilon_{-1}^{p,q}, \varepsilon_1^{p,q}\} \cup \mathcal{B}_{p, \lfloor \frac{p+q}{2} \rfloor} \cup \mathcal{B}_{\lfloor \frac{p+q}{2} \rfloor + 1, q} & p \neq q \end{cases}$$

and is analogous to a binary tree. We have the following approximation property for $\mathcal{B}_n := \mathcal{B}_{1,n}$,

**Proposition 3.** *The basis $\mathcal{B}_n$ is complete. Furthermore, for any labeling $\boldsymbol{u} \in \{-1, 1\}^n$ there exists a covering set $\mathcal{C}_{\boldsymbol{u}} \subseteq \mathcal{B}_n$ such that $|\mathcal{C}_{\boldsymbol{u}}| \leq 2(\Phi_{\mathcal{S}}(\boldsymbol{u}) + 1)\lceil \log_2 \frac{n}{2} \rceil$ for $n > 2$.*

From a computational perspective the binary tree structure ensures that there are only $\Theta(\log n)$ specialists active per trial, leading to an exponential speed-up in prediction. A similar set of specialists were used for obtaining adaptive-regret bounds in [9, 23] and data-compression in [33]. In those works however the "binary tree" structure is over the time dimension (trial sequence) whereas in this work the binary tree is over the space dimension (graph) and a fixed-share update is used to obtain adaptivity over the time dimension.[2]

In the corollary that follows we will exploit the fact that by making the algorithm *conservative* we may reduce the usual $\log T$ term in the mistake bound induced by a fixed-share update to $\log \log T$. A conservative algorithm only updates the specialists' weights on trials on which a mistake is made. Furthermore the bound given in the following corollary is *smooth* as the cost per switch will be measured with a Hamming-like divergence $H$ on the "cut" edges between successive labelings, defined as

$$H(\boldsymbol{u}, \boldsymbol{u}') := \sum_{(i,j) \in E_{\mathcal{S}}} [\, [[u_i \neq u_j] \vee [u_i' \neq u_j']] \wedge [[u_i \neq u_i'] \vee [u_j \neq u_j']] \,].$$

Observe that $H(\boldsymbol{u}, \boldsymbol{u}')$ is smaller than twice the hamming distance between $\boldsymbol{u}$ and $\boldsymbol{u}'$ and is often significantly smaller. To achieve the bounds we will need the following proposition, which upper bounds divergence $J$ by $H$, a subtlety is that there are many distinct sets of specialists consistent with a given comparator. For example, consider a uniform labeling on $\mathcal{S}$. One may "cover" this labeling with a single specialist or alternatively $n$ specialists, one covering each vertex. For the sake of simplicity in bounds we will always choose the smallest set of covering specialists. Thus we introduce the following formal definitions of *consistency* and *minimal-consistency*.

**Definition 4.** *A comparator $\boldsymbol{\mu} \in \Delta_{|\mathcal{E}|}$ is consistent with the labeling $\boldsymbol{u} \in \{-1,1\}^n$ if $\boldsymbol{\mu}$ is well-formed and $\mu_\varepsilon > 0$ implies that for all $v \in V$ that $\varepsilon(v) \in \{u_v, \square\}$.*

**Definition 5.** *A comparator $\boldsymbol{\mu} \in \Delta_{|\mathcal{E}|}$ is minimal-consistent with the labeling $\boldsymbol{u} \in \{-1,1\}^n$ if it is consistent with $\boldsymbol{u}$ and the cardinality of its support set $|\{\mu_\varepsilon : \mu_\varepsilon > 0\}|$ is the minimum of all comparators consistent with $\boldsymbol{u}$.*

**Proposition 6.** *For a linearized graph $\mathcal{S}$, for comparators $\boldsymbol{\mu}, \boldsymbol{\mu}' \in \Delta_{|\mathcal{F}_n|}$ that are minimal-consistent with $\boldsymbol{u}$ and $\boldsymbol{u}'$ respectively,*

$$J_{\mathcal{F}_n}(\boldsymbol{\mu}, \boldsymbol{\mu}') \leq \min\left(2H(\boldsymbol{u}, \boldsymbol{u}'), \Phi_{\mathcal{S}}(\boldsymbol{u}') + 1\right).$$

A proof is given in Appendix C. In the following corollary we summarize the results of the SCS algorithm using the basis sets $\mathcal{F}_n$ and $\mathcal{B}_n$ with an optimally-tuned switching parameter $\alpha$.

**Corollary 7.** *For a connected $n$-vertex graph $\mathcal{G}$ and with randomly sampled spine $\mathcal{S}$, the number of mistakes made in predicting the online sequence $(i_1, y_1), \ldots, (i_T, y_T)$ by the SCS algorithm with optimally-tuned $\alpha$ is upper bounded with basis $\mathcal{F}_n$ by*

$$\mathcal{O}\left(\Phi_1 \log n + \sum_{i=1}^{|K|-1} H(\boldsymbol{u}_{k_i}, \boldsymbol{u}_{k_{i+1}})\left(\log n + \log|K| + \log\log T\right)\right)$$

*and with basis $\mathcal{B}_n$ by*

$$\mathcal{O}\left(\left(\Phi_1 \log n + \sum_{i=1}^{|K|-1} H(\boldsymbol{u}_{k_i}, \boldsymbol{u}_{k_{i+1}})\left(\log n + \log|K| + \log\log T\right)\right)\log n\right)$$

*for any sequence of labelings $\boldsymbol{u}_1, \ldots, \boldsymbol{u}_T \in \{-1,1\}^n$ such that $u_{t,i_t} = y_t$ for all $t \in [T]$.*

Thus the bounds are equivalent up to a factor of $\log n$ although the computation times vary dramatically. See Appendix D for a technical proof of these results, and details on the selection of the switching parameter $\alpha$.

On the lower bound side, tight upper and lower bounds were proven for graph label prediction when the graph was a tree in [6]. We now give a sketch of a simple argument for a lower bound on the number of mistakes made for predicting a switching sequence of labelings on $\mathcal{S}$. We first describe how introducing and removing cuts can force mistakes in the simplest case.

Given a single graph-labeling problem on an unlabeled line graph $\mathcal{S}$, an adversary may force $\Theta(\log n)$ mistakes with a resultant cut-size $\Phi(\boldsymbol{u}) = 1$. In the switching case if $\mathcal{S}$ is uniformly labelled ($\Phi(\boldsymbol{u}) = 0$) and up to two cuts are introduced, then the `learner` can be forced to make $\mathcal{O}(\log n)$ mistakes. On the other hand if we have cut-size of $\Phi(\boldsymbol{u}') = 2$ an adversary when a "switch" occurs can force a single mistake with the outcome that the cut-size $\Phi(\boldsymbol{u}'') \in \{0, 1, 2\}$.

Now for a switching sequence of graph labelings, $\boldsymbol{u}_1, \ldots, \boldsymbol{u}_T$, let $\Phi(\boldsymbol{u}_t) \ll n$ for all $t$. For a labeling $\boldsymbol{u}$, $\mathcal{S}$ can be divided into $\Phi(\boldsymbol{u}) + 1$ segments of length $\frac{n}{\Phi(\boldsymbol{u})+1}$. Each segment can be made independent of one another by fixing the boundary vertices between segments. We therefore have $\Phi(\boldsymbol{u}) + 1$ independent learning problems and an adversary can force $\Theta(\log\left(\frac{n}{\Phi(\boldsymbol{u})}\right))$ mistakes for every two cuts introduced and 1 mistake for every 2 cuts removed.

While the bounds in Corollary 7 reflect the *smoothness* of the sequence of labelings, we pay $\mathcal{O}(\log n + \log|K| + \log\log T)$ for every cut removed *and* introduced for basis set $\mathcal{F}_n$, with an additional logarithmic factor for basis $\mathcal{B}_n$. There is therefore an interesting gap between these bounds and the sketched lower bound, not least of which caused by the $\log\log T$ term, which we conjecture should be possible to remove.

Table 1: Mean error $\pm$ std over 25 iterations on a 404-vertex graph for all algorithms and benchmarks, and for all ensemble sizes of SCS-F and SCS-B.

| Algorithm | Ensemble Size | | | | | | |
|---|---|---|---|---|---|---|---|
| | 1 | 3 | 5 | 9 | 17 | 33 | 65 |
| SCS-F | $1947 \pm 49$ | $1597 \pm 32$ | $1475 \pm 30$ | $1364 \pm 28$ | $1293 \pm 26$ | $1247 \pm 21$ | $1218 \pm 19$ |
| SCS-B | $1438 \pm 32$ | $1198 \pm 27$ | $1127 \pm 25$ | $1079 \pm 24$ | $1050 \pm 23$ | $1032 \pm 22$ | $1021 \pm 18$ |
| Kernel Perceptron | $3326 \pm 43$ | - | - | - | - | - | - |
| Local | $3411 \pm 55$ | - | - | - | - | - | - |
| Global | $4240 \pm 44$ | - | - | - | - | - | - |
| Temporal (Local) | $2733 \pm 42$ | - | - | - | - | - | - |
| Temporal (Global) | $3989 \pm 44$ | - | - | - | - | - | - |

Note that we may avoid the issue of needing to optimally tune $\alpha$ using the following method proposed by [14] and by [28]. We use a time-varying parameter and on trial $t$ we set $\alpha_t = \frac{1}{t+1}$. We have the following guarantee for this method, see Appendix E for a proof.

**Proposition 8.** *For a connected $n$-vertex graph $\mathcal{G}$ and with randomly sampled spine $\mathcal{S}$, the SCS algorithm with bases $\mathcal{F}_n$ and $\mathcal{B}_n$ in predicting the online sequence $(i_1, y_1), \ldots, (i_T, y_T)$ now with time-varying $\alpha$ set equal to $\frac{1}{t+1}$ on trial $t$ achieves the same asymptotic mistake bounds as in Corollary 7 with an optimally-tuned $\alpha$, under the assumption that $\Phi_{\mathcal{S}}(\boldsymbol{u}_1) \leq \sum_{i=1}^{|K|-1} J_{\mathcal{E}}(\boldsymbol{\mu}_{k_i}, \boldsymbol{\mu}_{k_{i+1}})$.*

## 4  Experiments

In this section we present results of experiments on real data. The City of Chicago currently contains 608 public bicycle stations for its "Divvy Bike" sharing system. Current and historical data is available from the City of Chicago[3] containing a variety of features for each station, including latitude, longitude, number of docks, number of operational docks, and number of docks occupied. The latest data on each station is published approximately every ten minutes.

We used a sample of 72 hours of data, consisting of three consecutive weekdays in April 2019. The first 24 hours of data were used for parameter selection, and the remaining 48 hours of data were used for evaluating performance. On each ten-minute snapshot we took the percentage of empty docks of each station. We created a binary labeling from this data by setting a threshold of $50\%$. Thus each bicycle station is a vertex in our graph and the label of each vertex indicates whether that station is 'mostly full' or 'mostly empty'. Due to this thresholding the labels of some 'quieter' stations were observed not to switch, as the percentage of available docks rarely changed. These stations tended to be on the 'outskirts', and thus we excluded these stations from our experiments, giving 404 vertices in our graph.

Using the geodesic distance between each station's latitude and longitudinal position a connected graph was built using the union of a $k$-nearest neighbor graph ($k = 3$) and a minimum spanning tree. For each instance of our algorithm the graph was then transformed in the manner described in Section 2.2, by first drawing a spanning tree uniformly at random and then linearizing using depth-first search.

As natural benchmarks for this setting we considered the following four methods. 1.) For all vertices predict with the most frequently occurring label of the entire graph from the training data ("Global"). 2.) For each vertex predict with its most frequently occurring label from the training data ("Local"). 3.) For all vertices at any given time predict with the most frequently occurring label of the entire graph at that time from the training data ("Temporal-Global") 4.) For each vertex at any given time predict with that vertex's label observed at the same time in the training data ("Temporal-Local"). We also compare our algorithms against a kernel Perceptron proposed by [18] for predicting switching graph labelings (see Appendix F for details).

Following the experiments of [7] in which ensembles of random spanning trees were drawn and aggregated by an unweighted majority vote, we tested the effect on performance of using ensembles of instances of our algorithms, aggregated in the same fashion. We tested ensemble sizes in $\{1, 3, 5, 9, 17, 33, 65\}$, using odd numbers to avoid ties.

For every ten-minute snapshot (labeling) we queried 30 vertices uniformly at random (with replacement) in an online fashion, giving a sequence of 8640 trials over 48 hours. The average performance

Figure 1: Left: Mean cumulative mistakes over 25 iterations for all algorithms and benchmarks over 48 hours (8640 trials) on a 404-vertex graph. A comparison of the mean performance of SCS with bases $\mathcal{F}_n$ and $\mathcal{B}_n$ (SCS-F and SCS-B respectively) using an ensemble of size 1 and 65 is shown. Right: An example of two binary labelings taken from the morning and evening of the first 24 hours of data. An 'orange' label implies that station is $< 50\%$ full and a 'black' label implies that station is $\geq 50\%$ full.

over 25 iterations is shown in Figure 1. There are several surprising observations to be made from our results. Firstly, both SCS algorithms performed significantly better than all benchmarks and competing algorithms. Additionally basis $\mathcal{B}_n$ outperformed basis $\mathcal{F}_n$ by quite a large margin, despite having the weaker bound and being exponentially faster. Finally we observed a significant increase in performance of both SCS algorithms by increasing the ensemble size (see Figure 1 and Table 1), additional details on these experiments and results of all ensemble sizes are given in Appendix G.

Interestingly when tuning $\alpha$ we found basis $\mathcal{B}_n$ to be very robust, while $\mathcal{F}_n$ was very sensitive. This observation combined with the logarithmic per-trial time complexity suggests that SCS with $\mathcal{B}_n$ has promise to be a very practical algorithm.

## 5   Conclusion

Our primary result was an algorithm for predicting switching graph labelings with a per-trial prediction time of $O(\log n)$ and a mistake bound that *smoothly* tracks changes to the graph labeling over time. In the long version of this paper we plan to extend the analysis of the primary algorithm to the expected regret setting by relaxing our simplifying assumption of the *well-formed* comparator sequence that is *minimal-consistent* with the labeling sequence. From a technical perspective the open problem that we found most intriguing is to eliminate the $\log \log T$ term from our bounds. The natural approach to this would be to replace the conservative *fixed-share* update with a *variable-share* update [21]; in our efforts however we found many technical problems with this approach. On both the more practical and speculative side; we observe that the specialists sets $\mathcal{B}_n$, and $\mathcal{F}_n$ were chosen to "prove bounds". In practice we can use any *hierarchical* graph clustering algorithm to produce a *complete* specialist set and furthermore multiple such clusterings may be pooled. Such a pooled set of subgraph "motifs" could be then be used for example in a multi-task setting (see for example, [27]).

## Footnotes

[1]However, see the analysis of WML in [29] for a precursory result.

[2]An interesting open problem is to try to find good bounds and time-complexity with sets of specialists over *both* the time and space dimensions.

[3]https://data.cityofchicago.org/Transportation/Divvy-Bicycle-Stations-Historical/eq45-8inv

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
