[Supplementary Material]

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

# Appendix

## A Proof of Theorem 2

*Proof.* Recall that the cached share update (1) is equivalent to performing (3). We thus simulate the latter update in our analysis. We first argue the inequality

$$[\hat{y}_t \neq y_t] \leq \frac{1}{\boldsymbol{\mu}_t(\mathcal{Y}_t)} \left( d(\boldsymbol{\mu}_t, \boldsymbol{\omega}_t) - d(\boldsymbol{\mu}_t, \dot{\boldsymbol{\omega}}_t) \right), \tag{5}$$

as this is derived by observing that

$$\begin{aligned}
d(\boldsymbol{\mu}_t, \boldsymbol{\omega}_t) - d(\boldsymbol{\mu}_t, \dot{\boldsymbol{\omega}}_t) &= \sum_{\varepsilon \in \mathcal{E}} \mu_{t,\varepsilon} \log \frac{\dot{\omega}_{t,\varepsilon}}{\omega_{t,\varepsilon}} \\
&= \sum_{\varepsilon \in \mathcal{Y}_t} \mu_{t,\varepsilon} \log \frac{\dot{\omega}_{t,\varepsilon}}{\omega_{t,\varepsilon}} \\
&\geq \boldsymbol{\mu}_t(\mathcal{Y}_t)[\hat{y}_t \neq y_t],
\end{aligned}$$

where the second line follows the fact that $\mu_{t,\varepsilon} \log \frac{\dot{\omega}_{t,\varepsilon}}{\omega_{t,\varepsilon}} = 0$ if $\varepsilon \notin \mathcal{Y}_t$ as either the specialist $\varepsilon$ predicts '□' and $\dot{\omega}_{t,\varepsilon} = \omega_{t,\varepsilon}$ or it predicts incorrectly and hence $\mu_{t,\varepsilon} = 0$. The third line follows as for $\varepsilon \in \mathcal{Y}_t$, $\frac{\dot{\omega}_{t,\varepsilon}}{\omega_{t,\varepsilon}} \geq 2$ if there has been a mistake on trial $t$ and otherwise the ratio is $\geq 1$. Indeed, since Algorithm 1 is conservative, this ratio is exactly 1 when no mistake is made on trial $t$, thus without loss of generality we will assume the algorithm makes a mistake on every trial.

For clarity we will now use simplified notation and let $\pi_t := \boldsymbol{\mu}_t(\mathcal{Y}_t)$. We now prove the following inequalities which we will add to (5) to create a telescoping sum of relative entropy terms and entropy terms.

$$\frac{1}{\pi_t} \left[ d(\boldsymbol{\mu}_t, \dot{\boldsymbol{\omega}}_t) - d(\boldsymbol{\mu}_t, \boldsymbol{\omega}_{t+1}) \right] \geq -\frac{1}{\pi_t} \log \frac{1}{1-\alpha}, \tag{6}$$

$$\frac{1}{\pi_t} d(\boldsymbol{\mu}_t, \boldsymbol{\omega}_{t+1}) - \frac{1}{\pi_{t+1}} d(\boldsymbol{\mu}_{t+1}, \boldsymbol{\omega}_{t+1}) \geq -\frac{1}{\pi_t} H(\boldsymbol{\mu}_t) + \frac{1}{\pi_{t+1}} H(\boldsymbol{\mu}_{t+1}) - J_{\mathcal{E}}(\boldsymbol{\mu}_t, \boldsymbol{\mu}_{t+1}) \log \frac{|\mathcal{E}|}{\alpha}. \tag{7}$$

Firstly (6) is proved with the following

$$d(\boldsymbol{\mu}_t, \dot{\boldsymbol{\omega}}_t) - d(\boldsymbol{\mu}_t, \boldsymbol{\omega}_{t+1}) = \sum_{\varepsilon \in \mathcal{E}} \mu_{t,\varepsilon} \log \frac{\omega_{t+1,\varepsilon}}{\dot{\omega}_{t,\varepsilon}} \geq \sum_{\varepsilon \in \mathcal{E}} \mu_{t,\varepsilon} \log \left( \frac{(1-\alpha)\dot{\omega}_{t,\varepsilon}}{\dot{\omega}_{t,\varepsilon}} \right) = \log(1-\alpha),$$

where the inequality has used $\omega_{t+1,\varepsilon} \geq (1-\alpha)\dot{\omega}_{t,\varepsilon}$ from (3).

To prove (7) we first define the following sets.

$$\begin{aligned}
\Theta_t &:= \{\varepsilon \in \mathcal{E} : \mu_{t-1,\varepsilon} \neq 0, \mu_{t,\varepsilon} = 0\}, \\
\Psi_t &:= \{\varepsilon \in \mathcal{E} : \mu_{t-1,\varepsilon} \neq 0, \mu_{t,\varepsilon} \neq 0\}, \\
\Omega_t &:= \{\varepsilon \in \mathcal{E} : \mu_{t-1,\varepsilon} = 0, \mu_{t,\varepsilon} \neq 0\}.
\end{aligned}$$

We now expand the following

$$\frac{1}{\pi_t}d(\boldsymbol{\mu}_t, \boldsymbol{\omega}_{t+1}) - \frac{1}{\pi_{t+1}}d(\boldsymbol{\mu}_{t+1}, \boldsymbol{\omega}_{t+1})$$

$$= \frac{1}{\pi_t}d(\boldsymbol{\mu}_t, \boldsymbol{\omega}_{t+1}) - \frac{1}{\pi_t}d(\boldsymbol{\mu}_{t+1}, \boldsymbol{\omega}_{t+1}) + \frac{1}{\pi_t}d(\boldsymbol{\mu}_{t+1}, \boldsymbol{\omega}_{t+1}) - \frac{1}{\pi_{t+1}}d(\boldsymbol{\mu}_{t+1}, \boldsymbol{\omega}_{t+1})$$

$$= \frac{1}{\pi_t}\sum_{\varepsilon \in \mathcal{E}}\mu_{t,\varepsilon}\log\frac{\mu_{t,\varepsilon}}{\omega_{t+1,\varepsilon}} - \frac{1}{\pi_t}\sum_{\varepsilon \in \mathcal{E}}\mu_{t+1,\varepsilon}\log\frac{\mu_{t+1,\varepsilon}}{\omega_{t+1,\varepsilon}}$$

$$+ \frac{1}{\pi_t}\sum_{\varepsilon \in \mathcal{E}}\mu_{t+1,\varepsilon}\log\frac{\mu_{t+1,\varepsilon}}{\omega_{t+1,\varepsilon}} - \frac{1}{\pi_{t+1}}\sum_{\varepsilon \in \mathcal{E}}\mu_{t+1,\varepsilon}\log\frac{\mu_{t+1,\varepsilon}}{\omega_{t+1,\varepsilon}}$$

$$= -\frac{1}{\pi_t}H(\boldsymbol{\mu}_t) + \frac{1}{\pi_t}H(\boldsymbol{\mu}_{t+1}) + \sum_{\varepsilon \in \mathcal{E}}\left(\frac{\mu_{t,\varepsilon}}{\pi_t} - \frac{\mu_{t+1,\varepsilon}}{\pi_t}\right)\log\frac{1}{\omega_{t+1,\varepsilon}}$$

$$- \frac{1}{\pi_t}H(\boldsymbol{\mu}_{t+1}) + \frac{1}{\pi_{t+1}}H(\boldsymbol{\mu}_{t+1}) + \sum_{\varepsilon \in \mathcal{E}}\left(\frac{\mu_{t+1,\varepsilon}}{\pi_t} - \frac{\mu_{t+1,\varepsilon}}{\pi_{t+1}}\right)\log\frac{1}{\omega_{t+1,\varepsilon}}. \qquad (8)$$

Recall that a comparator $\boldsymbol{\mu} \in \Delta_{|\mathcal{E}|}$ is *well-formed* if $\forall\, v \in V$, there exists a *unique* $\varepsilon \in \mathcal{E}$ such that $\varepsilon(v) \neq \square$ and $\mu_\varepsilon > 0$, and furthermore there exists a $\pi \in (0, 1]$ such that $\forall \varepsilon \in \mathcal{E} : \mu_\varepsilon \in \{0, \pi\}$, i.e., each specialist in the support of $\boldsymbol{\mu}$ has the same mass $\pi$ and these specialists disjointly cover the input space $(V)$. Thus, by collecting terms into the three sets $\Theta_{t+1}$, $\Psi_{t+1}$, and $\Omega_{t+1}$ we have

$$\sum_{\varepsilon \in \mathcal{E}}\left(\frac{\mu_{t,\varepsilon}}{\pi_t} - \frac{\mu_{t+1,\varepsilon}}{\pi_t}\right)\log\frac{1}{\omega_{t+1,\varepsilon}}$$

$$= \sum_{\varepsilon \in \Theta_{t+1}}\frac{\mu_{t,\varepsilon}}{\pi_t}\log\frac{1}{\omega_{t+1,\varepsilon}} + \sum_{\varepsilon \in \Psi_{t+1}}\left(\frac{\mu_{t,\varepsilon}}{\pi_t} - \frac{\mu_{t+1,\varepsilon}}{\pi_t}\right)\log\frac{1}{\omega_{t+1,\varepsilon}} - \sum_{\varepsilon \in \Omega_{t+1}}\frac{\mu_{t+1,\varepsilon}}{\pi_t}\log\frac{1}{\omega_{t+1,\varepsilon}}$$

$$= \sum_{\varepsilon \in \Theta_{t+1}}\frac{\mu_{t,\varepsilon}}{\pi_t}\log\frac{1}{\omega_{t+1,\varepsilon}} + \sum_{\varepsilon \in \Psi_{t+1}}\left(1 - \frac{\mu_{t+1,\varepsilon}}{\pi_t}\right)\log\frac{1}{\omega_{t+1,\varepsilon}} - \sum_{\varepsilon \in \Omega_{t+1}}\frac{\mu_{t+1,\varepsilon}}{\pi_t}\log\frac{1}{\omega_{t+1,\varepsilon}}, \qquad (9)$$

and similarly

$$\sum_{\varepsilon \in \mathcal{E}}\left(\frac{\mu_{t+1,\varepsilon}}{\pi_t} - \frac{\mu_{t+1,\varepsilon}}{\pi_{t+1}}\right)\log\frac{1}{\omega_{t+1,\varepsilon}}$$

$$= \sum_{\varepsilon \in \Psi_{t+1}}\left(\frac{\mu_{t+1,\varepsilon}}{\pi_t} - 1\right)\log\frac{1}{\omega_{t+1,\varepsilon}} + \sum_{\varepsilon \in \Omega_{t+1}}\left(\frac{\mu_{t+1,\varepsilon}}{\pi_t} - 1\right)\log\frac{1}{\omega_{t+1,\varepsilon}}. \qquad (10)$$

Substituting (9) and (10) into (8) and simplifying gives

$$\frac{1}{\pi_t}d(\boldsymbol{\mu}_t, \boldsymbol{\omega}_{t+1}) - \frac{1}{\pi_{t+1}}d(\boldsymbol{\mu}_{t+1}, \boldsymbol{\omega}_{t+1})$$

$$= -\frac{1}{\pi_t}H(\boldsymbol{\mu}_t) + \frac{1}{\pi_{t+1}}H(\boldsymbol{\mu}_{t+1}) + \sum_{\varepsilon \in \Theta_{t+1}}\frac{\mu_{t,\varepsilon}}{\pi_t}\log\frac{1}{\omega_{t+1,\varepsilon}} - \sum_{\varepsilon \in \Omega_{t+1}}\log\frac{1}{\omega_{t+1,\varepsilon}}$$

$$\geq -\frac{1}{\pi_t}H(\boldsymbol{\mu}_t) + \frac{1}{\pi_{t+1}}H(\boldsymbol{\mu}_{t+1}) - |\Omega_{t+1}|\log\frac{|\mathcal{E}|}{\alpha}, \qquad (11)$$

where the inequality has used the fact that $\frac{\alpha}{|\mathcal{E}|} \leq \omega_{t+1,\varepsilon} \leq 1$ from (3).

Summing over all trials then leaves a telescoping sum of relative entropy terms, a cost of $\frac{1}{\pi_t}\log\frac{1}{1-\alpha}$ on each trial, and $|\Omega_{t+1}|\log\frac{|\mathcal{E}|}{\alpha}$ for each switch. Thus,

$$\sum_{t=1}^{T}[\hat{y}_t \neq y_t] \leq \frac{1}{\pi_1}d(\boldsymbol{\mu}_1, \boldsymbol{\omega}_1) + \frac{1}{\pi_1}H(\boldsymbol{\mu}_1) + \sum_{t=1}^{T}\frac{1}{\pi_t}\log\frac{1}{1-\alpha} + \sum_{i=1}^{|K|-1}J_{\mathcal{E}}\left(\boldsymbol{\mu}_{k_i}, \boldsymbol{\mu}_{k_{i+1}}\right)\log\frac{|\mathcal{E}|}{\alpha}, \qquad (12)$$

where $J_{\mathcal{E}}(\boldsymbol{\mu}_{k_i}, \boldsymbol{\mu}_{k_{i+1}}) = |\Omega_{k_{i+1}}|$, and since $\boldsymbol{\omega}_1 = \frac{1}{|\mathcal{E}|}\mathbf{1}$, we can combine the remaining entropy and relative entropy terms to give $\frac{1}{\pi_1}d(\boldsymbol{\mu}_1, \boldsymbol{\omega}_1) + \frac{1}{\pi_1}H(\boldsymbol{\mu}_1) = \frac{1}{\pi_1}\log|\mathcal{E}|$, concluding the proof. $\qquad\square$

# B  Proof of Proposition 3

We recall the proposition:

*The basis $\mathcal{B}_n$ is complete. Furthermore, for any labeling $\boldsymbol{u} \in \{-1,1\}^n$ there exists a covering set $\mathcal{C}_{\boldsymbol{u}} \subseteq \mathcal{B}_n$ such that $|\mathcal{C}_{\boldsymbol{u}}| \leq 2(\Phi_{\mathcal{S}}(\boldsymbol{u}) + 1)\lceil \log_2 \frac{n}{2} \rceil$.*

We first give a brief intuition of the proof; any required terms will be defined more completely later. For a given labeling $\boldsymbol{u} \in \{-1,1\}^n$ of cut-size $\Phi_{\mathcal{S}}(\boldsymbol{u})$, the spine $\mathcal{S}$ can be cut into $\Phi_{\mathcal{S}}(\boldsymbol{u}) + 1$ *clusters*, where a cluster is a contiguous segment of vertices with the same label. We will upper bound the maximum number of cluster specialists required to cover a single cluster, and therefore obtain an upper bound for $|\mathcal{C}_{\boldsymbol{u}}|$ by summing over the $\Phi_{\mathcal{S}}(\boldsymbol{u}) + 1$ clusters.

Without loss of generality we assume $n = 2^r$ for some integer $r$ and thus the structure of $\mathcal{B}_n$ is analogous to a perfect binary tree of depth $d = \log_2 n$. Indeed, for a fixed label parameter $y$ we will adopt the terminology of binary trees such that for instance we say specialist $\varepsilon_y^{i,j}$ for $i \neq j$ has a so-called *left-child* $\varepsilon_y^{i,\lfloor \frac{i+j}{2} \rfloor}$ and *right-child* $\varepsilon_y^{\lceil \frac{i+j}{2} \rceil, j}$. Similarly, we say that $\varepsilon_y^{i,\lfloor \frac{i+j}{2} \rfloor}$ and $\varepsilon_y^{\lceil \frac{i+j}{2} \rceil, j}$ are *siblings*, and $\varepsilon_y^{i,j}$ is their *parent*. Note that any specialist is both an ancestor and a descendant of itself, and a proper descendant of a specialist is a descendant of one of its children. Finally the *depth* of specialist $\varepsilon_y^{i,j}$ is defined to be equal to the depth of the corresponding node in a binary tree, such that $\varepsilon_y^{1,n}$ is of depth 0, $\varepsilon_y^{1,\frac{n}{2}}$ and $\varepsilon_y^{\frac{n}{2}+1,n}$ are of depth 1, etc.

The first claim of the proposition is easy to prove as $\{\varepsilon_{-1}^{i,i}, \varepsilon_1^{i,i} : i \in [n]\} \subset \mathcal{B}_n$ and thus any labeling $\boldsymbol{u} \in \{-1,1\}^n$ can be covered. We now prove the second claim of the proposition.

We will denote a uniformly-labeled contiguous segment of vertices by the pair $(l,r)$, where $l, r \in [n]$ are the two end vertices of the segment. For completeness we will allow the trivial case when $l = r$. Given a labeling $\boldsymbol{u} \in \{-1,1\}^n$, let $\mathcal{L}_{\boldsymbol{u}} := \{(l,r) : 1 \leq l \leq r \leq n; u_l = \ldots = u_r; u_{l-1} \neq u_l; u_{r+1} \neq u_r\}$ be the set of maximum-sized contiguous segments of unifmormly-labeled vertices. Note that $u_{l-1}$ or $u_{r+1}$ may be vacuous. When the context is clear, we will also describe $(l,r)$ as a *cluster*, and as the set of vertices $\{l, \ldots, r\}$.

For a given $\boldsymbol{u} \in \{-1,1\}^n$ and cluster $(l,r) \in \mathcal{L}_{\boldsymbol{u}}$, we say $\mathcal{B}_{(l,r)} \subseteq \mathcal{B}_n$ is an $(l,r)$-covering set with respect to $\boldsymbol{u}$ if for all $\varepsilon_y^{i,j} \in \mathcal{B}_{(l,r)}$ we have $l \leq i, j \leq r$, and if for all $k \in (l,r)$ there exists some $\varepsilon_y^{i,j} \in \mathcal{B}_{(l,r)}$ such that $i \leq k \leq j$ and $y = u_k$. That is, every vertex in the cluster is 'covered' by at least one specialist and no specialists cover any vertices $k \notin (l,r)$. We define $\mathcal{D}^{(l,r)}$ to be the set of all possible $(l,r)$-covering sets with respect to $\boldsymbol{u}$.

We now define
$$\delta(\mathcal{B}_{(l,r)}) := |\mathcal{B}_{(l,r)}|$$
to be the *complexity* of $\mathcal{B}_{(l,r)} \in \mathcal{D}^{(l,r)}$.

For a given $\boldsymbol{u} \in \{-1,1\}^n$ and cluster $(l,r) \in \mathcal{L}_{\boldsymbol{u}}$, we wish to produce an $(l,r)$-covering set of *minimum* complexity, which we denote $\mathcal{B}_{(l,r)}^* := \underset{\mathcal{B}_{(l,r)} \in \mathcal{D}^{(l,r)}}{\operatorname{argmin}} \delta(\mathcal{B}_{(l,r)})$. Note that an $(l,r)$-covering set of minimum complexity cannot contain any two specialists which are siblings, since they can be removed from the set and replaced by their parent specialist.

**Lemma 9.** *For any $\boldsymbol{u} \in \{-1,1\}^n$, for any $(l,r) \in \mathcal{L}_{\boldsymbol{u}}$, the $(l,r)$-covering set of minimum complexity, $\mathcal{B}_{(l,r)}^* = \underset{\mathcal{B}_{(l,r)} \in \mathcal{D}^{(l,r)}}{\operatorname{argmin}} \delta(\mathcal{B}_{(l,r)})$ contains at most two specialists of each unique depth.*

*Proof.* We first give an intuitive sketch of the proof. For a given $\boldsymbol{u} \in \{-1,1\}^n$ and cluster $(l,r) \in \mathcal{L}_{\boldsymbol{u}}$ assume that there are at least three specialists of equal depth in $\mathcal{B}_{(l,r)}^*$, then any of these specialists that are in the 'middle' may be removed, along with any of their siblings or proper descendants that are also members of $\mathcal{B}_{(l,r)}^*$ without creating any 'holes' in the covering, decreasing the complexity of $\mathcal{B}_{(l,r)}^*$.

We use a proof by contradiction. Suppose for contradiction that for a given $\boldsymbol{u} \in \{-1,1\}^n$ and $(l,r) \in \mathcal{L}_{\boldsymbol{u}}$, the $(l,r)$-covering set of minimum complexity, $\mathcal{B}_{(l,r)}^*$, contains three distinct specialists of the same depth, $\varepsilon_y^{a,b}, \varepsilon_y^{c,d}, \varepsilon_y^{e,f}$. Without loss of generality let $a, b < c, d < e, f$. Note that we have

$l \leq a < f \leq r$. We consider the following two possible scenarios: when two of the three specialists are siblings, and when none are.

If $\varepsilon_y^{a,b}$ and $\varepsilon_y^{c,d}$ are siblings, then we have $\varepsilon_y^{a,d} \in \mathcal{B}_n$ and thus $\{\varepsilon_y^{a,d}\} \cup \mathcal{B}_{(l,r)}^* \setminus \{\varepsilon_y^{a,b}, \varepsilon_y^{c,d}\}$ is an $(l,r)$-covering set of smaller complexity, leading to a contradiction. The equivalent argument holds if $\varepsilon_y^{c,d}$ and $\varepsilon_y^{e,f}$ are siblings.

If none are siblings, then let $\varepsilon_y^{c',d'}$ be the sibling of $\varepsilon_y^{c,d}$ and let $\varepsilon_y^{C,D}$ be the parent of $\varepsilon_y^{c,d}$ and $\varepsilon_y^{c',d'}$. Note that $a,b < c',d',c,d$ and $c',d',c,d < e,f$ and hence $l < C < D < r$. If an ancestor of $\varepsilon_y^{C,D}$ is in $\mathcal{B}_{(l,r)}^*$, then $\mathcal{B}_{(l,r)}^* \setminus \{\varepsilon_y^{c,d}\}$ is an $(l,r)$-covering set of smaller complexity, leading to a contradiction. Alternatively, if no ancestor of $\varepsilon_y^{C,D}$ is in $\mathcal{B}_{(l,r)}^*$, then $\varepsilon_y^{c',d'}$ or some of its proper descendants must be in $\mathcal{B}_{(l,r)}^*$, otherwise there exists some vertex $k' \in (c',d')$ such that there exists no specialist $\varepsilon_y^{i,j} \in \mathcal{B}_{(l,r)}^*$ such that $i \leq k' \leq j$, and therefore $\mathcal{B}_{(l,r)}^*$ would not be an $(l,r)$-covering set. Let $\varepsilon_y^{p,q}$ be a descendant of $\varepsilon_y^{c',d'}$ which is contained in $\mathcal{B}_{(l,r)}^*$. Then $\{\varepsilon_y^{C,D}\} \cup \mathcal{B}_{(l,r)}^* \setminus \{\varepsilon_y^{c,d}, \varepsilon_y^{p,q}\}$ is an $(l,r)$-covering set of smaller complexity, leading to a contradiction.

We conclude that there can be no more than 2 specialists of the same depth in $\mathcal{B}_{(l,r)}^*$ for any $\boldsymbol{u} \in \{-1,1\}^n$ and any $(l,r) \in \mathcal{L}_{\boldsymbol{u}}$. $\qquad\square$

We now prove an upper bound on the maximum minimum-complexity of an $(l,r)$-covering set under any labeling $\boldsymbol{u}$.

**Corollary 10.** *For all* $\boldsymbol{u} \in \{-1,1\}^n$,

$$\max_{(l,r)\in\mathcal{L}_{\boldsymbol{u}}} \min_{\mathcal{B}_{(l,r)}\in\mathcal{D}^{(l,r)}} \delta(\mathcal{B}_{(l,r)}) \leq 2\log\frac{n}{2}\,. \tag{13}$$

*Proof.* For any $\boldsymbol{u} \in \{-1,1\}^n$, and $(l,r) \in \mathcal{L}_{\boldsymbol{u}}$, since $\mathcal{B}_{(l,r)}^*$ can contain at most 2 specialists of the same depth (Lemma 9) an $(l,r)$-covering set of minimum-complexity can have at most two specialists of depths $2, 3, \ldots, d$. This cannot contain two specialists of depth 1 as they are siblings. This upper bounds the maximum minimum-complexity of any $(l,r)$-covering set by $2(d-2) = 2\log\frac{n}{2}$. $\qquad\square$

Finally we conclude that for any labeling $\boldsymbol{u} \in \{-1,1\}^n$ of cut-size $\Phi_{\mathcal{S}}(\boldsymbol{u})$, there exists $\mathcal{C}_{\boldsymbol{u}} \subseteq \mathcal{B}_n$ such that $|\mathcal{C}_{\boldsymbol{u}}| \leq 2\log_2\left(\frac{n}{2}\right)(\Phi_{\mathcal{S}}(\boldsymbol{u})+1)$.

# C   Proof of Proposition 6

First recall the proposition statement.

**Proposition 11.** *For a linearized graph $\mathcal{S}$, for comparators $\boldsymbol{\mu}, \boldsymbol{\mu}' \in \Delta_{|\mathcal{F}_n|}$ that are minimal-consistent with $\boldsymbol{u}$ and $\boldsymbol{u}'$ respectively,*

$$J_{\mathcal{F}_n}(\boldsymbol{\mu}, \boldsymbol{\mu}') \leq \min\left(2H(\boldsymbol{u}, \boldsymbol{u}'), \Phi_{\mathcal{S}}(\boldsymbol{u}')+1\right).$$

*Proof.* We prove both inequalities separately. We first prove $J_{\mathcal{F}_n}(\boldsymbol{\mu}, \boldsymbol{\mu}') \leq \Phi_{\mathcal{S}}(\boldsymbol{u}')+1$. This follows directly from the fact that $J_{\mathcal{E}}(\boldsymbol{\mu}, \boldsymbol{\mu}') := |\{\varepsilon \in \mathcal{E} : \mu_\varepsilon = 0, \mu'_\varepsilon \neq 0\}|$ and therefore $J_{\mathcal{F}_n}(\boldsymbol{\mu}, \boldsymbol{\mu}') \leq |\{\varepsilon \in \mathcal{F}_n : \mu'_\varepsilon \neq 0\}| = \Phi_{\mathcal{S}}(\boldsymbol{u}')+1$.

We now prove $J_{\mathcal{F}_n}(\boldsymbol{\mu}, \boldsymbol{\mu}') \leq 2H(\boldsymbol{u}, \boldsymbol{u}')$. Recall that if $\boldsymbol{u} \neq \boldsymbol{u}'$ then by definition of the minimal-consistent comparators $\boldsymbol{\mu}$ and $\boldsymbol{\mu}'$, the set $\{\varepsilon \in \mathcal{F}_n : \mu_\varepsilon = 0, \mu'_\varepsilon \neq 0\}$ corresponds to the set of maximum-sized contiguous segments of vertices in $\mathcal{S}$ sharing the same label in the labeling $\boldsymbol{u}'$ that did not exist in the labeling $\boldsymbol{u}$. From here on we refer to a maximum-sized contiguous segment as just a contiguous segment.

When switching from labeling $\boldsymbol{u}$ to $\boldsymbol{u}'$, we consider the following three cases. First when a non-cut edge (with respect to $\boldsymbol{u}$) becomes a cut edge (with respect to $\boldsymbol{u}'$), second when a cut edge (with respect to $\boldsymbol{u}$) becomes a non-cut edge (with respect to $\boldsymbol{u}'$), and lastly when a cut edge remains a cut edge, but the labeling of the two corresponding vertices are 'swapped'.

Formally then, for an edge $(i,j) \in E_\mathcal{S}$ such that $[u_i = u_j] \wedge [u'_i \neq u'_j]$ there exists two new contiguous segments of vertices sharing the same label that did not exist in the labeling $\boldsymbol{u}$, their boundary being the edge $(i,j)$.

Conversely for an edge $(i,j) \in E_\mathcal{S}$ such that $[u_i \neq u_j] \wedge [u'_i = u'_j]$ there exists one new contiguous segment of vertices sharing the same label that did not exist in the labeling $\boldsymbol{u}$, that segment will contain the edge $(i,j)$.

Finally for an edge $(i,j) \in E_\mathcal{S}$ such that $[[u_i \neq u_j] \wedge [u'_i \neq u'_j]] \wedge [[u_i \neq u'_i] \vee [u_j \neq u'_j]]$ there exists two new contiguous segments of vertices sharing the same label that did not exist in the labeling $\boldsymbol{u}$, their boundary being the edge $(i,j)$.

We conclude that the number of new contiguous segments of vertices sharing the same label that did not exist in the labeling $\boldsymbol{u}$ is upper bounded by

$$2 \sum_{(i,j) \in E_\mathcal{S}} [[u_i \neq u_j] \vee [u'_i \neq u'_j]] \wedge [[u_i \neq u'_i] \vee [u_j \neq u'_j]].$$

$\square$

## D  Proof of Corollary 7

First recall the corollary statement.

**Corollary 11.** *For a connected $n$-vertex graph $\mathcal{G}$ and with randomly sampled spine $\mathcal{S}$, the number of mistakes made in predicting the online sequence $(i_1, y_1), \ldots, (i_T, y_T)$ by the SCS algorithm with optimally-tuned $\alpha$ is upper bounded with basis $\mathcal{F}_n$ by*

$$\mathcal{O}\left( \Phi_1 \log n + \sum_{i=1}^{|K|-1} H(\boldsymbol{u}_{k_i}, \boldsymbol{u}_{k_{i+1}})\left(\log n + \log |K| + \log \log T\right) \right)$$

*and with basis $\mathcal{B}_n$ by*

$$\mathcal{O}\left( \left( \Phi_1 \log n + \sum_{i=i}^{|K|-1} H(\boldsymbol{u}_{k_i}, \boldsymbol{u}_{k_{i+1}})\left(\log n + \log |K| + \log \log T\right) \right) \log n \right)$$

*for any sequence of labelings $\boldsymbol{u}_1, \ldots, \boldsymbol{u}_T \in \{-1, 1\}^n$ such that $u_{t,i_t} = y_t$ for all $t \in [T]$.*

*Proof.* Since Algorithm 1 has a conservative update, we may ignore trials on which no mistake is made and thus from the point of view of the algorithm a mistake is made on every trial, we will therefore assume that $T = M$. This will lead to a self-referential mistake bound in terms of the number of mistakes made which we will then iteratively substitute into itself.

Let $c := \log_2 e$, we will use the fact that $\log_2\left(\frac{1}{1-\frac{x}{y+x}}\right) \leq c\frac{x}{y}$ for $x, y > 0$. We will first optimally tune $\alpha$ to give our tuned mistake bound for a general basis set $\mathcal{E}$, and then derive the bounds for bases $\mathcal{F}_n$ and $\mathcal{B}_n$ respectively. The value of $\alpha$ that minimizes (4) is

$$\alpha = \frac{\sum\limits_{i=1}^{|K|-1} J_\mathcal{E}\left(\boldsymbol{\mu}_{k_i}, \boldsymbol{\mu}_{k_{i+1}}\right)}{\sum\limits_{t=1}^{T} \frac{1}{\pi_t} + \sum\limits_{i=1}^{|K|-1} J_\mathcal{E}\left(\boldsymbol{\mu}_{k_i}, \boldsymbol{\mu}_{k_{i+1}}\right)}, \tag{14}$$

which when substituted into the second term of (4) gives

$$M_\mathcal{E} \leq \frac{1}{\pi_1} \log |\mathcal{E}| + c \sum_{i=1}^{|K|-1} J_\mathcal{E}\left(\boldsymbol{\mu}_{k_i}, \boldsymbol{\mu}_{k_{i+1}}\right) + \sum_{i=1}^{|K|-1} J_\mathcal{E}\left(\boldsymbol{\mu}_{k_i}, \boldsymbol{\mu}_{k_{i+1}}\right) \log \frac{|\mathcal{E}|}{\alpha}. \tag{15}$$

We now upper bound $\frac{1}{\alpha}$ for substitution in the last term of (15) for bases $\mathcal{F}_n$ and $\mathcal{B}_n$ separately.

**Basis $\mathcal{F}_n$ :** For $\mathcal{F}_n$ observe that $|\mathcal{E}| = n^2 + n$, and since any labeling $\boldsymbol{u}_t \in \{-1, 1\}^n$ of cut-size $\Phi_{\mathcal{S}}(\boldsymbol{u}_t)$ is covered by $\Phi_{\mathcal{S}}(\boldsymbol{u}_t) + 1$ specialists, we have that $\pi_t = 1/(\Phi_{\mathcal{S}}(\boldsymbol{u}_t) + 1)$ on all trials. We let the number of mistakes made by SCS with basis $\mathcal{F}_n$ be denoted by $M_{\mathcal{F}_n}$. Thus (15) immediately becomes

$$M_{\mathcal{F}_n} \leq (\Phi_1 + 1) \log |\mathcal{F}_n| + c \sum_{i=1}^{|K|-1} J_{\mathcal{F}_n}(\boldsymbol{\mu}_{k_i}, \boldsymbol{\mu}_{k_{i+1}}) + \sum_{i=1}^{|K|-1} J_{\mathcal{F}_n}(\boldsymbol{\mu}_{k_i}, \boldsymbol{\mu}_{k_{i+1}}) \log \frac{|\mathcal{F}_n|}{\alpha}. \quad (16)$$

To upper bound $\frac{1}{\alpha}$ we note that if $\boldsymbol{\mu}_{k_i} \neq \boldsymbol{\mu}_{k_{i+1}}$ then $J_{\mathcal{F}_n}(\boldsymbol{\mu}_{k_i}, \boldsymbol{\mu}_{k_{i+1}}) \geq 1$, and that for $\mathcal{F}_n$, $\frac{1}{\pi_i} = \Phi_{k_i} + 1 \leq n$, thus from (14) we have

$$\frac{1}{\alpha} = 1 + \frac{\sum_{t=1}^{T} \frac{1}{\pi_t}}{\sum_{i=1}^{|K|-1} J_{\mathcal{F}_n}(\boldsymbol{\mu}_{k_i}, \boldsymbol{\mu}_{k_{i+1}})} \leq 1 + \frac{nT}{|K| - 1} \leq \frac{nT + |K| - 1}{|K| - 1} \leq \frac{(n+1)T}{|K| - 1}.$$

Substituting $\frac{1}{\alpha} \leq \frac{(n+1)T}{|K|-1}$ into (16) gives

$$M_{\mathcal{F}_n} \leq (\Phi_1 + 1) \log |\mathcal{F}_n| + \sum_{i=1}^{|K|-1} J_{\mathcal{F}_n}(\boldsymbol{\mu}_{k_i}, \boldsymbol{\mu}_{k_{i+1}}) \left[ \log(e|\mathcal{F}_n|) + \log(n+1) + \log \frac{T}{|K|-1} \right] \quad (17)$$

We now show our method to reduce the $\log T$ term in our bound to $\log \log T$ by substituting the self-referential mistake bound into itself. We first simplify (17) and substitute $T = M_{\mathcal{F}_n}$,

$$M_{\mathcal{F}_n} \leq \underbrace{(\Phi_1 + 1) \log |\mathcal{F}_n| + \sum_{i=1}^{|K|-1} J_{\mathcal{F}_n}(\boldsymbol{\mu}_{k_i}, \boldsymbol{\mu}_{k_{i+1}}) \log \left( \frac{e|\mathcal{F}_n|(n+1)}{|K|-1} \right)}_{=:\mathcal{Z}}$$

$$+ \underbrace{\sum_{i=1}^{|K|-1} J_{\mathcal{F}_n}(\boldsymbol{\mu}_{k_i}, \boldsymbol{\mu}_{k_{i+1}})}_{=:\mathcal{J}} \log M_{\mathcal{F}_n}$$

$$\leq \mathcal{Z} + \mathcal{J} \log(\mathcal{Z} + \mathcal{J} \log M_{\mathcal{F}_n})$$

$$\leq \mathcal{Z} + \mathcal{J} \log \mathcal{Z} + \mathcal{J} \log \mathcal{J} + \mathcal{J} \log \log M_{\mathcal{F}_n},$$

using $\log(a + b) \leq \log(a) + \log(b)$ for $a, b \geq 2$. We finally use the fact that $\mathcal{J} = \mathcal{O}(n|K|)$ to give $\mathcal{J} \log \mathcal{J} = \mathcal{O}(\mathcal{J} \log(n|K|))$ and similarly

$$\mathcal{J} \log \mathcal{Z} = \mathcal{O}(\mathcal{J} \log(\Phi_1 \log n + \mathcal{J} \log n))$$
$$= \mathcal{O}(\mathcal{J} \log((n + \mathcal{J}) \log n)))$$
$$= \mathcal{O}(\mathcal{J} \log(n + \mathcal{J}))$$
$$= \mathcal{O}(\mathcal{J} \log(n|K|)),$$

to give

$$M_{\mathcal{F}_n} \leq \mathcal{O}\left( \Phi_1 \log n + \sum_{i=1}^{|K|-1} J_{\mathcal{F}_n}(\boldsymbol{\mu}_{k_i}, \boldsymbol{\mu}_{k_{i+1}})(\log n + \log |K| + \log \log T) \right).$$

**Basis $\mathcal{B}_n$:** For $\mathcal{B}_n$ we apply the same technique as above, but first observe the following. Without loss of generality assume $n = 2^r$ for some integer $r$, we then have $|\mathcal{E}| = 4n - 2$. We let the number of mistakes made by SCS with basis $\mathcal{B}_n$ be denoted by $M_{\mathcal{B}_n}$. Thus for basis $\mathcal{B}_n$ (15) becomes

$$M_{\mathcal{B}_n} \leq 2 \log \frac{n}{2} (\Phi_1 + 1) \log |\mathcal{B}_n| + c \sum_{i=1}^{|K|-1} J_{\mathcal{B}_n}(\boldsymbol{\mu}_{k_i}, \boldsymbol{\mu}_{k_{i+1}}) + \sum_{i=1}^{|K|-1} J_{\mathcal{B}_n}(\boldsymbol{\mu}_{k_i}, \boldsymbol{\mu}_{k_{i+1}}) \log \frac{|\mathcal{B}_n|}{\alpha}. \quad (18)$$

Recall proposition 3 (that $|\mathcal{C}_{\boldsymbol{u}}| \leq 2\log_2\left(\frac{n}{2}\right)(\Phi_{\mathcal{S}}(\boldsymbol{u})+1)$) and since $\pi_t = \frac{1}{|\mathcal{C}_{\boldsymbol{u}}|}$, then for any labeling $\boldsymbol{u}_t \in \{-1,1\}^n$ of cut-size $\Phi_{\mathcal{S}}(\boldsymbol{u}_t)$ we have $\frac{1}{2(\Phi_{\mathcal{S}}(\boldsymbol{u}_t)+1)\log\frac{n}{2}} \leq \pi_t \leq \frac{1}{\Phi_{\mathcal{S}}(\boldsymbol{u}_t)+1}$. We then apply the same argument upper bounding $\frac{1}{\alpha}$,

$$\frac{1}{\alpha} = 1 + \frac{\sum_{t=1}^{T}\frac{1}{\pi_t}}{\sum_{i=1}^{|K|-1} J_{\mathcal{B}_n}\big(\boldsymbol{\mu}_{k_i},\boldsymbol{\mu}_{k_{i+1}}\big)}$$
$$\leq 1 + \frac{2n\log\left(\frac{n}{2}\right)T}{|K|-1} \leq \frac{2n\log\left(\frac{n}{2}\right)T + |K|-1}{|K|-1} \leq \frac{\left(2n\log\left(\frac{n}{2}\right)+1\right)T}{|K|-1},$$

and substituting $\frac{1}{\alpha} \leq \frac{(2n\log\left(\frac{n}{2}\right)+1)T}{|K|-1}$ into the last term of (18) gives

$$M_{\mathcal{B}_n} \leq 2\log_2\frac{n}{2}\left(\Phi_1 + 1\right)\log|\mathcal{B}_n| +$$
$$\sum_{i=1}^{|K|-1} J_{\mathcal{B}_n}\big(\boldsymbol{\mu}_{k_i},\boldsymbol{\mu}_{k_{i+1}}\big)\left[c + \log|\mathcal{B}_n| + \ln 2n + \log\frac{T}{|K|-1} + \log\log n\right].$$

Applying the same recursive technique as above yields a bound of

$$M_{\mathcal{B}_n} \leq \mathcal{O}\left(\Phi_1\left(\log n\right)^2 + \sum_{i=1}^{|K|-1} J_{\mathcal{B}_n}\big(\boldsymbol{\mu}_{k_i},\boldsymbol{\mu}_{k_{i+1}}\big)\left(\log n + \log|K| + \log\log T\right)\right).$$

Using the same argument given in proposition 3 for any two labelings $\boldsymbol{u},\boldsymbol{u}' \in \{-1,1\}^n$, for two consistent well-formed comparators $\boldsymbol{\mu},\boldsymbol{\mu}' \in \Delta_{|\mathcal{B}_n|}$ respectively, and for two consistent well-formed comparators $\hat{\boldsymbol{\mu}},\hat{\boldsymbol{\mu}}' \in \Delta_{|\mathcal{F}_n|}$, we have that $J_{\mathcal{B}_n}(\boldsymbol{\mu},\boldsymbol{\mu}') \leq 2\log\frac{n}{2}J_{\mathcal{F}_n}(\hat{\boldsymbol{\mu}},\hat{\boldsymbol{\mu}}')$. Finally we use $J_{\mathcal{F}_n} \leq 2H(\boldsymbol{u},\boldsymbol{u}')$ from Proposition 6 to complete the proof. $\qquad\square$

## E    Proof of Proposition 8

*Proof.* Using a time-dependent $\alpha$ we can re-write (4) as

$$M_{\mathcal{E}} \leq \frac{1}{\pi_1}\log|\mathcal{E}| + \sum_{t=1}^{T}\frac{1}{\pi_t}\log\frac{1}{1-\alpha_t} + \sum_{i=1}^{|K|-1} J_{\mathcal{E}}\big(\boldsymbol{\mu}_{k_i},\boldsymbol{\mu}_{k_{i+1}}\big)\log\frac{|\mathcal{E}|}{\alpha_{k_{i+1}}}, \tag{19}$$

and letting $\alpha_t := \frac{1}{t+1}$, and letting $c := \log_2 e$, gives the following,

$$M_{\mathcal{E}} \leq \frac{1}{\pi_1}\log|\mathcal{E}| + \sum_{t=1}^{T}\frac{1}{\pi_t}\log\frac{1}{1-\frac{1}{t+1}} + \sum_{i=1}^{|K|-1} J_{\mathcal{E}}\big(\boldsymbol{\mu}_{k_i},\boldsymbol{\mu}_{k_{i+1}}\big)\log\left(|\mathcal{E}|\left(k_{i+1}+1\right)\right) \tag{20}$$

$$\leq \frac{1}{\pi_1}\log|\mathcal{E}| + c\sum_{t=1}^{T}\frac{1}{\pi_t}\frac{1}{t} + \sum_{i=1}^{|K|-1} J_{\mathcal{E}}\big(\boldsymbol{\mu}_{k_i},\boldsymbol{\mu}_{k_{i+1}}\big)\log\left(|\mathcal{E}|T\right) \tag{21}$$

$$\leq \frac{1}{\pi_1}\log|\mathcal{E}| + c\left(\max_{t\in[T]}\frac{1}{\pi_t}\right)\sum_{t=1}^{T}\frac{1}{t} + \sum_{i=1}^{|K|-1} J_{\mathcal{E}}\big(\boldsymbol{\mu}_{k_i},\boldsymbol{\mu}_{k_{i+1}}\big)\log\left(|\mathcal{E}|T\right) \tag{22}$$

$$\leq \frac{1}{\pi_1}\log|\mathcal{E}| + \left(\max_{t\in[T]}\frac{1}{\pi_t}\right)\log\left(eT\right) + \sum_{i=1}^{|K|-1} J_{\mathcal{E}}\big(\boldsymbol{\mu}_{k_i},\boldsymbol{\mu}_{k_{i+1}}\big)\log\left(|\mathcal{E}|T\right) \tag{23}$$

where the step from (20) to (21) has used $\log_2\left(1+x\right) \leq cx$ for $x > 0$, and the step from (22) to (23) has used $\sum_{t\in[T]}\frac{1}{t} < \int_1^T\frac{1}{t}dt + 1 = \ln\left(eT\right) = \frac{1}{c}\log_2\left(eT\right)$.

We now use the following upper bound on $\max\limits_{t\in[T]}\frac{1}{\pi_t}$,

$$\max_{t\in[T]}\frac{1}{\pi_t} \leq \frac{1}{\pi_1} + \sum_{i=1}^{|K|-1} J_{\mathcal{E}}(\boldsymbol{\mu}_{k_i},\boldsymbol{\mu}_{k_{i+1}}),$$

and the assumption that $\sum_{i=1}^{|K|-1} J_{\mathcal{E}}(\boldsymbol{\mu}_{k_i},\boldsymbol{\mu}_{k_{i+1}}) \geq \frac{1}{\pi_1}$, to give

$$\max_{t\in[T]}\frac{1}{\pi_t} \leq 2 \sum_{i=1}^{|K|-1} J_{\mathcal{E}}(\boldsymbol{\mu}_{k_i},\boldsymbol{\mu}_{k_{i+1}}). \tag{24}$$

Substituting (24) into (23) then gives

$$M_{\mathcal{E}} \leq \frac{1}{\pi_1}\log|\mathcal{E}| + 2\sum_{i=1}^{|K|-1} J_{\mathcal{E}}\left(\boldsymbol{\mu}_{k_i},\boldsymbol{\mu}_{k_{i+1}}\right)\left(\log(eT) + \frac{1}{2}\log(|\mathcal{E}|T)\right)$$

$$= \frac{1}{\pi_1}\log|\mathcal{E}| + 2\sum_{i=1}^{|K|-1} J_{\mathcal{E}}\left(\boldsymbol{\mu}_{k_i},\boldsymbol{\mu}_{k_{i+1}}\right)\left(\frac{1}{2}\log(|\mathcal{E}|) + \log(e) + \frac{3}{2}\log(T)\right)$$

Using a conservative update (see section 3.1), we similarly set $\alpha_t := \frac{1}{m+1}$, where $m$ is the current number of mistakes of the algorithm. We next use the same 'recursive trick' as that in the proof of Corollary 7. The proof follows analogously, leaving

$$M_{\mathcal{F}_n} \leq \mathcal{O}\left(\Phi_1\log n + \sum_{i=1}^{|K|-1} J_{\mathcal{F}_n}(\boldsymbol{\mu}_{k_i},\boldsymbol{\mu}_{k_{i+1}})(\log n + \log|K| + \log\log T)\right)$$

for the basis set $\mathcal{F}_n$, and

$$M_{\mathcal{B}_n} \leq \mathcal{O}\left(\Phi_1(\log n)^2 + \sum_{i=1}^{|K|-1} J_{\mathcal{B}_n}(\boldsymbol{\mu}_{k_i},\boldsymbol{\mu}_{k_{i+1}})(\log n + \log|K| + \log\log T)\right)$$

for the basis set $\mathcal{B}_n$. $\qquad\square$

## F  The Switching Graph Perceptron

In this section for completeness we provide the kernelized Perceptron algorithm for switching graph prediction. The algorithm is described and a mistake bound given for the switching-graph labeling problem in [18, Sec. 6.2].

The key to the approach is to use the following graph kernel (introduced by [19]) $\boldsymbol{K} := \boldsymbol{L}_{\mathcal{G}}^+ + R_L\mathbf{1}\mathbf{1}^\top$ with $R_L := \max_i(\boldsymbol{e}_i^\top \boldsymbol{L}_{\mathcal{G}}^+ \boldsymbol{e}_i)$, where $\boldsymbol{L}_{\mathcal{G}}^+$ denotes the pseudo-inverse of the graph Laplacian, and for $i \in [n]$, we let $\boldsymbol{e}_i$ denote the $i$-th unit basis vector, i.e., $e_{i,i'} = 0$ if $i \neq i'$ and equals 1 if $i' = i$. The norm induced by this kernel is denoted $\|\boldsymbol{u}\|_{\boldsymbol{K}} := \sqrt{\boldsymbol{u}^\top \boldsymbol{K}^{-1} \boldsymbol{u}}$.

## G  Further Details on Experiments

In this section we give further details on the experimental methods of Section 4. Data was collected spanning 72 hours from $4:55am$ on $8^{th}$ April 2019 to $4:55am$ on $11^{th}$ April 2019. Any stations that were not in service during any of the 72 hours were removed (in this case there was only one such station).

As described in Section 4, the variable measured was the percentage of occupied docks in each station, and a threshold of $50\%$ was set to induce a binary labeling. Any stations whose induced labeling did not change over the 72 hours were also removed from the dataset. This left a graph of 404 stations.

The first 24 hours of data were used for parameter selection. Parameters were tuned using exhaustive search over the ranges specified in Table 2, taking the mean minimizer over 10 iterations.

**input** : Graph $\mathcal{G}$
**parameter** : $\gamma > 0$
**initialize** : $\boldsymbol{w}_1 \leftarrow \boldsymbol{0}$
$\boldsymbol{K} \leftarrow \boldsymbol{L}_{\mathcal{G}}^{+} + \max_{i \in [n]} (\boldsymbol{e}_i^{\top} \boldsymbol{L}_{\mathcal{G}}^{+} \boldsymbol{e}_i) \boldsymbol{1}\boldsymbol{1}^{\top}$
**for** $t = 1$ **to** $T$ **do**
    **receive** $i_t \in V$
    **predict** $\hat{y}_t \leftarrow sign(w_{t,i_t})$
    **receive** $y_t \in \{-1, 1\}$
    **if** $\hat{y}_t \neq y_t$ **then**
        $\dot{\boldsymbol{w}}_t \leftarrow \boldsymbol{w}_t + y_t \frac{\boldsymbol{K}\boldsymbol{e}_{i_t}}{\boldsymbol{K}_{i_t,i_t}}$
        **if** $\|\dot{\boldsymbol{w}}_t\|_{\boldsymbol{K}} > \gamma$ **then**
            $\boldsymbol{w}_{t+1} \leftarrow \frac{\dot{\boldsymbol{w}}_t}{\|\dot{\boldsymbol{w}}_t\|_{\boldsymbol{K}}}\gamma$
        **else**
            $\boldsymbol{w}_{t+1} \leftarrow \dot{\boldsymbol{w}}_t$

**Algorithm 2:** SWITCHING GRAPH PERCEPTRON

Table 2: Parameter ranges used for optimizing the three algorithms with tunable parameters.

| Algorithm | Parameter | Parameter Range | Optimized Parameter |
|---|---|---|---|
| Kernel Perceptron | $\gamma$ | $3.5 - 5$ | $3.89$ |
| SCS-F | $\alpha$ | $1 \times 10^{-12} - 1 \times 10^{-6}$ | $7.4 \times 10^{-10}$ |
| SCS-B | $\alpha$ | $1 \times 10^{-5} - 5 \times 10^{-4}$ | $3.0 \times 10^{-4}$ |