[Reviews · NeurIPS 2019]

Reviewer 1



The authors of this submission present an algorithm for online prediction of graph labelings (in fact switching labelings). Their algorithm is based on using standard techniques from the literature to obtain a mistakes bound whose main component is a non symmetric cost of switching that takes into account only experts entering the comparator sequence. The main original result of the paper seems to be in section 3.1., where the construction of the specialists and their updates and runtime is analyzed and the binary tree basis is introduced. It appears this is the main idea behind the improvements claimed in the paper.

Reviewer 2



This paper studies the problem of online prediction of binary labels in a graph. Nature first chooses a sequence of labelings of the graph and at each round, the learner receives a vertex, predicts its label and receives the true label. The labels may vary over time and the goal of the learner is to minimize his number of mistakes. The paper proposes an algorithm based on sampling a random spanning tree, linearizing it and combining a well-chosen basis of experts that predict constant predictions over clusters of vertices. An upper-bound on the expected number of mistakes is provided which depends on the regularity of the labelings over the graph and over time. Experiments on real data are finally provided with convincing results. Strengths: - The clarity and quality of the submission are ok. The results are interesting and the experiments are convincing. - I liked the fact that the upper-bounds smoothly depend on the variation of the labelings and that the labelings of the entire graph could be arbitrary given that they match the observed labels. Weaknesses: - To my opinion, the setting and the algorithm lack a bit of originality and might seem as incremental combinations of methods of graph labelings prediction and online learning in a switching environment. Yet, the algorithm for graph labelings is efficient, new and seem different from the existing ones. - Lower bounds and optimality of the results are not discussed. In the conclusion section, it is asked whether the loglog(T) can be removed. Does this mean that up to this term the bounds are tight? I would like more discussions on this. More comparison with existing upper-bounds and lower-bound without switches could be made for instance. In addition, this could be interesting to plot the upper-bound on the experiments, to see how tight is the analysis. Other comments: - Only bounds in expectation are provided. Would it be possible to get high-probability bounds? For instance by using ensemble methods as performed in the experiments. Some measure about the robustness could be added to the experiments (such as error bars or standard deviation) in addition to the mean error. - When reading the introduction, I thought that the labels were adversarially chosen by an adaptive adversary. It seems that the analysis is only valid when all labels are chosen in advance by an oblivious adversary. Am I right? This should maybe be clarified. - This paper deals with many graph notions and it is a bit hard to get into it but the writing is generally good though more details could sometimes be provided (definition of the resistance distance, more explanations on Alg. 1 with brief sentences defining A_t, Y_t,...). - How was alpha tuned in the experiments (as 1/(t+1) or optimally)? - Some possible extensions could be discussed (are they straightforward?): directed or weighted graph, regression problem (e.g, to predict the number of bikes in your experiment)... Typo: l 268: the sum should start at 1

Reviewer 3



The authors introduce a practical O(log n) per step online algorithm for predicting graph labelings by a novel combination/appropriate adaptation of online learning techniques and provide a mistake bound analysis and some experimental support. The paper is well written and clearly presented, and does a good job of surveying related work and putting their method into context. A clear accept. Minor comments: - line 239: earlier work [Partition Tree Weighting, Veness, White, Bowling, Gyogy, DCC, 2013] also used a similar binary temporal discretization to provide a O(log n) algorithm for non-stationary time series modelling.

[Author Response · NeurIPS 2019]

We would like to thank the reviewers for their insightful and helpful comments. We address some specific points raised below, and would like to thank the reviewers in advance for considering our responses.

**Reviewer** #3

Thank you for your comments.

**Reviewer** #4

*To my opinion, the optimality and the tightness of the results should be discussed* - We fully agree, a discussion was initially omitted due to space constraints. First note, in terms of the prior literature for graph labeling without switching lower bounds were proved in references [6,7]. Lower bounds for switching in the experts model are given in [1,21]. These are for the "log" and "mix" loss respectively, these are both unbounded loss functions compared to "mistake" counting. For these unbounded loss functions there is a $\Omega(\Phi \log(T/\Phi))$ term in their lower bound but as we speculate in line 328 in our model the dependence on $T$ may be an artifact. A sketch of a lower bound on a line graph is provided below if desired this can be summarised into a theorem and included in the appendix if suggested by the reviewers.

Observe that if we have a single graph-labeling problem on an $n$-vertex line graph with a cut size of 1 it is not difficult to force $O(\log n)$ mistakes; likewise if we have a uniformly labeled line graph hence no cuts and a single cut is introduced we can force $O(\log n)$ mistakes. On the other hand if we have a line graph with a cut size $\Phi$ we can force $\Phi/4$ mistakes by "removing" $\Phi/2$ cuts. Now for a switching sequence of graph labeling problems, $\boldsymbol{\mu}_1, \ldots, \boldsymbol{\mu}_T$, let $\Phi(\boldsymbol{\mu}_t) \ll n$ for all $t$. For a labeling $\boldsymbol{\mu}$ observe that we can divide the line graph into $\Phi(\boldsymbol{\mu}) + 1$ segments, of length $\frac{n-1}{\Phi(\boldsymbol{\mu})+1}$, where each segment can be made independent of one another by fixing the boundary vertices between segments. We therefore have $\Phi(\boldsymbol{\mu}) + 1$ independent learning problems and can force $\Theta(\log \frac{n}{\Phi(\boldsymbol{\mu})})$ mistakes for every cut introduced and 1 mistake whenever we remove 2 cuts. Note beside the $\log \log T$ dependence there still remains some gap between this sketched lower bound and Corollary 7.

*Might seem as incremental combinations of graph prediction and online learning* - We believe that a logarithmic-time adaptive online algorithm for graph prediction is a significant improvement over the state of the art. Furthermore the bound in Theorem 2 is a new result for switching in the specialists setting with the asymmetric J() appearing in the bound.

*How was $\alpha$ tuned in the experiments?* - Lines 286/622: $\alpha$ was tuned using exhaustive search over the ranges given in the appendices over 24 hours of training data, rather than optimally from the bound.

*Adaptive vs. oblivious adversary* - All formally numbered theorems, corollaries and lemmas as stated are correct with an **adaptive** adversary. However, if as discussed in lines 152-157, one would like to convert the *deterministic* mistake bounds with respect to the *cut* to *expected* mistake bounds with respect to the *resistance weighted cut* then it is sufficient for that conversion to assume an **oblivious** adversary. We can add this additional comment to lines 152-157.

*Would it be possible to get high-probability bounds?* We suspect the technique in lines 152-157 could be adapted to a high probability mistake bound.

*268: the sum should start at 1* - Thank you for pointing that out.

*Could be interesting to plot the upper bound on the experiments* - We are happy to plot the computed upper bounds for the given data on the experiments, thank you for the suggestion.

*Measure of robustness (error bars/standard deviations)* - Error bars were initially omitted for 'neatness' in the plot. The standard deviations given in Table 2 in the appendix do imply an increasing robustness with larger ensemble sizes. We are happy to include a discussion on this and include error bars in the plot.

*More explanations on Alg. 1 with brief sentences defining $A_t$, $Y_t$.* The notations $A_t$, $Y_t$ are defined within the Algorithm, at their first appearance. Resistance distance (effective resistance) is defined lines 53-54, we can also supply the equation,

$$r(i,j) = \frac{1}{\min_{\boldsymbol{u} \in \Re^n} \sum_{(p,q) \in E} (u_p - u_q)^2 : u_i - u_j = 1} \, .$$

**Reviewer** #5

Thank you for your comments. In particular thank you for the additional reference we will add to line 239.

[Meta-Review · NeurIPS 2019]

This is a clear accept: all reviewers liked the paper, and I agree with their recommendation, as the paper provides a nice combination of fixed share (with specialists) with graph predictions. The authors are encouraged to include the lower bound. Also, the strength of the paper could be emphasized very clearly by comparing to applying meta-algorithms, such as those of [12-or rather its journal version, 13, 23] (these algorithms are specifically equipped to combine tracking with a large structured predictor class, at the price of a log T increase in complexity). Finally, I'd like to mention that two of the three reviewers were experts in proving regret bounds.